# HCV-HIV Chronic Coinfection Prevalence in Amazon Region

**DOI:** 10.3390/jcm11247284

**Published:** 2022-12-08

**Authors:** Marcio Nahum Lobo, Susan Denice Flores Irias, Pedro Leão Fontes Neto, Maria Eduarda Sousa Avelino, Maria Karoliny da Silva Torres, Marlinda de Carvalho Souza, Ricardo Roberto Souza Fonseca, Pedro Eduardo Bonfim Freitas, Heloisa Marceliano Nunes, José Raul Rocha de Araújo Júnior, Dickson Ciro Nascimento de Brito, Aldemir Branco Oliveira-Filho, Luiz Fernando Almeida Machado

**Affiliations:** 1Biology of Infectious and Parasitic Agents Post-Graduate Program, Federal University of Pará, Belém 66075-110, PA, Brazil; 2Virology Laboratory, Institute of Biological Sciences, Federal University of Pará, Belém 66075-110, PA, Brazil; 3Evandro Chagas Institute, Health Ministry of Brazil, Ananindeua 67030-000, PA, Brazil; 4Study and Research Group on Vulnerable Populations, Institute for Coastal Studies, Federal University of Pará, Bragança 68600-000, PA, Brazil

**Keywords:** hepatitis C virus (HCV), HIV, chronic infection, epidemiology, coinfection

## Abstract

Hepatitis C virus (HCV) infection is an important public health problem, especially in areas with a low human development index such as the Amazon region. This study aimed to identify the prevalence and genotypes of HCV among people living with HIV (PLWH), both neglected chronic diseases in the Amazon region. From March 2016 to June 2017, 433 PWLH were attended to at two sexually transmitted infection referral centers in the city of Belém, in the Brazilian state of Pará in the Amazon region. All individuals were submitted to testing via the rapid immunochromatographic assay (RIA) for the qualitative detection of anti-HCV antibodies. Samples with anti-HCV antibodies were evaluated by reverse transcriptase polymerase chain reaction (RT-PCR), and samples with HCV RNA were subjected to nucleotide sequencing and phylogenetic analysis. Three (0.7%) PLWH had anti-HCV antibodies, and only one (0.2%) had HCV RNA (genotype 2); of these, 31 (7.1%) self-declared to have used drugs at least one time, and 12 (2.7%) regularly use injected drugs. One participant was elderly, single, heterosexual, with a history of unprotected sex and multiple sexual partners. This study detected a low prevalence of HCV infection and recorded the presence of HCV genotype 2 for the first time among PLWH in the Brazilian Amazon.

## 1. Introduction

In 2022, according to the World Health Organization (WHO), the hepatitis C virus (HCV) is a global public health issue and a prevalent pathogen in acute and chronic viral hepatitis infection, with worldwide prevalence estimated to be at 58 million people infected with HCV and about 1.5 million new HCV infections per year. In 2019, around 290,000 people infected with HCV died from hepatic cirrhosis and hepatocellular carcinoma [1]. HCV incidence is classified by regions, and as claimed by Messina et al. [2] and Shah et al. [3], the highest incidence of cases is in the Eastern Mediterranean Region, European Region, Southeast Asian Region, Western Pacific Region, African Region, and the American Region.

HCV global prevalence and distribution might be explained by its morphology, transmission, and genotypes. HCV is a single-stranded RNA genome virus that belongs to the *Flaviviridae* family and the Hepacivirus genus. Its genome is approximately 9.6 kb in length and encodes a polyprotein of 3000 amino acid residues, which is cleaved to form structural and nonstructural viral proteins [4]. Due to these characteristics, during its replication, the HCV viral RNA polymerase has an increased tendency to produce errors throughout its continuous replications, which then yields high nucleotide substitution rates and a vast genotypic variability. Therefore, HCV is classified into 7 genotypes and 67 subgenotypes based on sequence data, and both genotypes and subgenotypes show specific global distribution; for instance, HCV genotype 1 has the highest global prevalence, followed by genotypes 3, 4, 2, 5, 6, and HCV genotype 7, which was isolated in Canada in a single case [4,5,6].

In Brazil, HCV national distribution, diagnosis, treatment, and vaccination are regulated by the Brazilian Ministry of Health (BMH) and the National Health Unique System (NHUS) Brazil, and as reported by BMH [7,8] in 2022, annual national HCV prevalence was highest in the Southeast region with 58.9%, followed by the South region with 27.5%, the Northeast region with 6.5%, the Midwest region with 3.6%, and the North region with 3.5%. Based on genotypes, Nutini et al. [9] identified HCV genotype 1 to be the most prevalent in the Brazilian territory with 71.1% of cases, followed by genotypes 3, 2, 4, and 5. In the northern region, HCV genotypes 1 (75.3%) and 3 (24.7%) were the most prevalent, as demonstrated by Castro et al. [4], and although HCV genotype 2 had already been discovered [10], an HCV subgenotype 2 study was not yet published, especially in relation to HCV-HIV coinfection.

In recent years, several studies have demonstrated that HCV-HIV coinfection is a relevant association in the clinical treatment of both HIV and HCV because this is a common coinfection due to the similarity in their transmission routes [5,9,11,12,13,14,15]. The existing literature has observed that HCV-HIV coinfection leads to a dynamic and rapid interaction between these two viruses because HCV/HIV coinfection can exacerbate the decrease in CD4+ T cell counts and overload the host’s immune system, thus increasing morbidity or producing a higher risk of severe hepatic cirrhosis, severe liver fibrosis, and hepatocellular carcinoma. This was the case among Brazilian risk group populations, mainly in hemodialysis patients, people who used drugs (PWUDs), prison populations, and people living with HIV (PLWH).

Thus, HCV diagnosis as well as HCV genotype and subgenotype identification might be of great importance in public health policy-making and can significantly contribute to a better quality of life among HCV-infected patients, mostly those with HIV coinfection, as a better treatment for both infections can be developed. Therefore, this study aimed to estimate the prevalence of HCV infections and identify the HCV genotypes among PLWH in the city of Belém in northern Brazil.

## 2. Materials and Methods

### 2.1. Type of Study and Sample Collection

This descriptive, cross-sectional study is based on serological data from PLWH attended to at two sexually transmitted infection (STIs) referral centers located in the city of Belém, Pará, northern Brazil (Figure 1). All individuals attended to in those two referral centers during the period between March 2016 and June 2017 were invited to participate in the study; 542 potential individuals later signed the written informed consent form after being selected according the following inclusion criteria: (a) age ≥ 18 years, (b) confirmed HIV infection, (c) agreed to participate in the study, and (d) signed the free and informed consent form. The exclusion criteria were patients with cognitive impairment who were unable to answer the questionnaire in an appropriate way.

### 2.2. Ethics

Written informed consent was obtained from all individuals for the publication of potentially identifiable images or data included in this article. This study was approved by the Committee for Ethics in Research of the Research by the Health Sciences Institute of the Federal University of Pará in Pará (PA), Brazil (protocol number: 2.601.161).

### 2.3. HCV Diagnosis

Peripheral 10 mL blood samples from all PLWH were collected and analyzed by RIA to qualitatively detect anti-HCV antibodies (Alere HCV^®^ Standard Diagnostics, Inc., Seoul, Republic of Korea). The samples that were nonreactive were classified as HCV-negative, and no additional tests were performed on those samples. The samples that were reactive in the anti-HCV rapid test were analyzed for the presence of anti-HCV antibodies using EIA (DIA.PRO, Diagnostic Bioprobes Milano, Italy), and were subsequently subjected to RT-PCR (AMPLICOR^®^, Roche Molecular Systems, Brachburg, NJ, USA) to confirm HCV infection. All the laboratory tests were performed according to the manufacturer’s guidelines.

### 2.4. Genotyping and Phylogenetic Analysis

Only positive samples with HCV RNA were subjected to nested PCR to amplify a 400 bp fragment from the non-structural region 5B (NS5B) gene, including part of the RNA-dependent RNA polymerase, using S and AS primers and specific PCR conditions that have previously been described [16,17]. The PCR products were subjected to 1.5% agarose gel electrophoresis and subsequently purified using the QIAquick PCR Purification Kit (QIAGEN). Nucleotide sequencing was performed using the Big Dye Terminator 3.1 kit (Applied Biosystems, Waltham, MA, USA) by capillary electrophoresis in the system (ABI PRISM 3500, Applied Biosystems). The sequence was edited and aligned using the AliView software [18] and evaluated by geno2pheno (hcv) (https://hcv.geno2pheno.org, accessed on 21 March 2022) [19]. 

In addition, a phylogenetic tree was reconstructed using an alignment containing nucleotide sequences of the HCV NS5B gene deposited in GenBank (Genotype 1: JQ323442, GQ379771, HQ630388, JQ323471, JQ323477, JQ323480; Genotype 2: AB047639, AB030907, JX227966, AB031663, JX227967; Genotype 3: JQ323492, JQ323497, JQ323501, KP324190, KP324192, KJ470616; Genotype 4: JX227976, JX227977, JX227978; Genotype 5: AY373484, AY373486, AY373487; Genotype 6: AY878650, AY878651, AY878652) and sequences obtained in this study. A maximum likelihood phylogenetic tree was reconstructed with the use of PhyML 3.1 (http://www.atgc-montpellier.fr/phyml/, accessed on 21 March 2021) under the best nucleotide substitution model, selected using the Smart Model Selection software integrated into the PhyML Web server [20,21]. The heuristic tree search was performed using the SPR branch-swapping algorithm, and the branch support was calculated with the approximate likelihood ratio (aLRT) SH-like test. The tree was drawn with FigTree 1.4.4 (http://tree.bio.ed.ac.uk/software/figtree/, accessed on 25 March 2021). The presence of resistance-associated substitutions (RASs) was also assessed using the online tools geno2pheno (hcv) and HCV-GLUE (http://hcv-glue.cvr.gla.ac.uk/#/home, accessed on 31 November 2022) [22]. The sequence obtained in this study was deposited in GenBank (MZ853084).

## 3. Results

From March 2016 to June 2017, 542 potential participants were attended to at two STI referral centers in the Brazilian city of Belém. Of these, 109 were excluded for not being infected with HIV or not having confirmed HIV infection. Thus, the sample for this study consisted of 433 PLWH. The average age was 33.5 years (36.2% were over 40 years old). Most of the PLWH were male (64.7%), heterosexual (62.4%), and single (69.3%), had completed high school (47.3%), used condoms during intercourse (46.4%), had a single partner (63.7%), and 31 (7.1%) self-declared to have used drugs at least one time, and 12 (2.7%) regularly use injected drugs.

In total, three (3/433–0.7%) tested positive for anti-HCV antibodies using the rapid immunochromatographic assay (RIA) and enzyme immunoassay (EIA). Among these three samples, only one (1/433–0.2%) tested positive for HCV RNA using reverse transcriptase polymerase chain reaction (RT-PCR). This participant with an active HCV infection was male, 63 years old, heterosexual, single, had tested positive for anti-HIV antibodies for more than five years, had a history of sex with female sex workers, sometimes used a condom during sex, and reported to never have used intravenous drugs (Table 1). This participant also reported to have engaged in unprotected sex and had had more than 10 sexual partners in the previous 12 months, with some partners coming from other Brazilian states. As shown in Figure 2, HCV genotype 2b was identified. There was no disagreement in the diagnosis of the HCV genotype provided by the online tool and the phylogenetic reconstruction. No RASs were identified, and susceptibility to sofosbuvir was indicated.

## 4. Discussion

The present study identified a single positive case, with the presence of anti-HCV genotype 2b antibodies, among HCV-HIV infected individuals in northern Brazil. To the best of the authors’ knowledge, this is the first paper in the literature to report such a case, especially in northern Brazil, which is an unusual region for the presence of the HCV genotype 2b. HCV infection varies through innumerous clinical characteristics, geographic location, socioeconomic parameters, exposure risk behaviors such as drug use, or medical conditions such as patients who have undergone hemodialysis due to chronic renal failure. Most epidemiological studies on HCV monoinfection or HIV coinfection that have identified genotype 2 were conducted in the southern and south-eastern regions of Brazil, mainly in patients who had undergone hemodialysis [23,24]. The lower amount of studies conducted in northern Brazil may be the result of geographic scarcity and the lack of effort by governmental and non-governmental institutions to detect and care for people infected with HCV in that region.

However, in the city of Belém, Freitas et al. [25] conducted a study in patients who had undergone hemodialysis to evaluate the prevalence of HCV antibodies. The authors found that 8.4% were anti-HCV positive, and HCV genotyping prevalence was 86.1% for genotype 1, 11.6% for genotype 2, and 2.3% for genotype 3. Sawada et al. [10] also conducted a study on 312 HCV-infected individuals who were attended to at a private hemodialysis clinic in Belém, and HCV genotype 1 (90.1%) was the most prevalent, followed by HCV genotype 2 (3.3%) and HCV genotype 3 (6.6%). Concerning these results, it may be possible to conclude that HCV genotype 2 is associated with hemodialysis. Besides patients who had undergone hemodialysis, other risk groups were evaluated in the state of Pará; Oliveira-Filho et al. [26] investigated the prevalence of HCV monoinfection among people using non-injecting drugs such as cannabis or cocaine paste, and HCV monoinfection prevalence was 28%, while the most prevalent HCV genotypes were HCV genotype 1 (76.9%) and HCV genotype 3 (23.1%).

In the Marajó archipelago in Breves city, Pacheco et al. [27] evaluated the prevalence of HCV monoinfection among people using injecting drugs (PUID) and found that the prevalence of anti-HCV antibodies was 36.9%; however, no HCV genotyping was performed. Similarly, in the Marajó archipelago, Silva et al. [28] investigated PUID and found that 28.3% had anti-HCV antibodies and 25.5% had HCV-RNA; HCV genotypes 1 and 3 were detected, but there was no evidence of the presence of HCV genotype 2. In Belém city, Oliveira-Filho et al. [29] evaluated the prevalence of HCV monoinfection among 1666 PUID; of these, 577 (34.6%) had HCV antibodies, 384 (23.1%) had active HCV infection, and HCV genotypes 1 and 3 were detected. In Belém city, the prevalence of HCV monoinfection among blood donors was investigated, and HCV infection prevalence was found to be low (0.13%), although HCV genotypes 1 and 3 were highly detected [4,30]. Based on these results, HCV genotypes 1 and 3 appear to be more prevalent among blood donors and PWUDs; however, in patients who had undergone hemodialysis, HCV genotype 2 seems to be more prevalent [31].

Although other studies have utilized HCV genotyping among key populations to identify HCV monoinfection in northern Brazil, this study presents interesting findings because for the first time, HCV-HIV coinfection and its genotyping are investigated. Unfortunately, HCV subgenotype 2b was positive in just one case; our hypothesis for such low prevalence is based on the high prevalence of HCV genotype 2 on the African continent, in the Caribbean regions, or in countries such as Indonesia and Vietnam. However, HCV genotype 2 is also frequent in European countries such as the Netherlands or on the American continent due to the transatlantic slave trade, which might explain why HCV genotype 2 seems to be uncommonly present in northern Brazil, because this region has the majority of indigenous populations or people of European descent [32]. Despite the low prevalence of HCV genotype 2 in HCV-HIV coinfection, Brazilian national authorities must carefully monitor HCV coinfection because the prevalence of HCV infections among PLWH in northern Brazil increased (from 2.5% to 2.7%) from 2009 to 2019 and in order to prevent new possibilities of new coinfections similar to COVID-19.

Therefore, epidemiological studies on HCV in key populations should be conducted in the future, mainly to plan actions and strategies aimed at performing diagnoses, vaccination, providing access to antiviral medications, and preventing HCV infections. These are essential to reduce the spread of HCV and other pathogens in the Brazilian population—especially in vulnerable groups such as PLWH in northern Brazil, which historically have difficulties in gaining access to HCV antiviral medications—and to avoid uncontrolled antiviral use in order to prevent drug resistance. The absence of RASs and susceptibility to sofosbuvir are other findings that reinforce the good prognosis for PLWH with HCV genotype 2. In Brazil, treatment for hepatitis C is offered to all people co-infected with HIV free of charge by the Ministry of Health, including a recommendation for the preferential use of sofosbuvir + daclatasvir on Brazilians co-infected with HIV-HCV due to the lower probability of drug interactions with antiretrovirals [33,34]. 

Sofosbuvir-based regimens for the treatment of HCV genotype 2 are safe and well-tolerated by patients [35]. Despite these findings, this study has limitations as the prevalence of HCV infection among PLWH recorded here was obtained from participants attended to at two STI referral centers. This epidemiological report did not clinically assess the study subjects. NS5B clonal sequencing or next-generation sequencing has not been performed in this study, so it is still not safe to exclude the possibility of known RASs, and we could also have missed hepatitis C infection during the window period prior to antibodies being detected by the screening tests.

## 5. Conclusions

Finally, the prevalence of HCV infection among PLWH is low in one of the main cities in the Brazilian Amazon, and the presence of HCV subgenotype 2b increases the importance of the epidemiological surveillance of HCV-HIV coinfection.

## Figures and Tables

**Figure 1 jcm-11-07284-f001:**
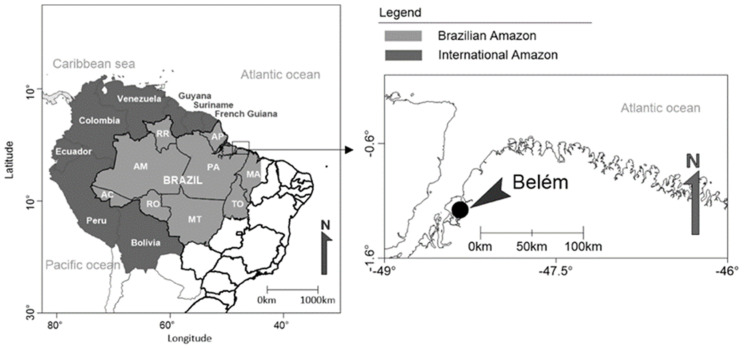
Geographic location of the city of Belém, in the Brazilian state of Pará (PA), Amazon region. Belém is the second largest city in the Brazilian Amazon.

**Figure 2 jcm-11-07284-f002:**
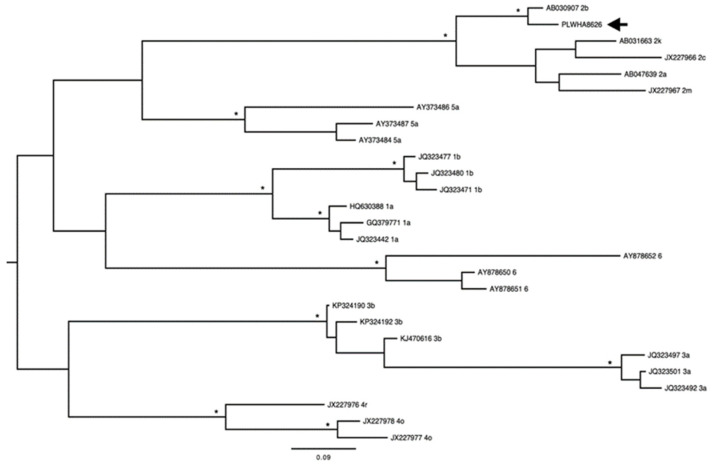
Maximum likelihood phylogenetic tree of hepatitis C virus (HCV) nucleotide sequences (from non-structural region 5B gene (313 nt)) isolated in a person living with HIV/AIDS in the Brazilian city of Belém, Amazon region, with other sequences of this hepatotropic virus deposited on GenBank. The tree was rooted at the midpoint. The asterisks point to key nodes with high support (SH-aLRT scores ≥ 0.90). The sample of this study can be identified by the acronym PLWH + number and is h (the arrow is to identify the analyzed area).

**Table 1 jcm-11-07284-t001:** Positive HCV genotype 2b patient’s medical information.

Patient Information	
Age (years)	63 years
Sex	Male
HIV diagnosed (years)	7 years (2010)
Sexual orientation	Heterosexual
Marital status	Single
Smoking status	Occasionally
Drinking status	Occasionally
Illegal drugs	Never used
HIV-related medications	Tenofovir + Lamivudine + Efavirenz (300 mg + 300 mg + 600 mg/day)
Laboratory values	
CD4 T lymphocyte (cells/mm^3^)	4.610 cells/mm^3^
Leukocytes count (cells/mm^3^)	846 cells/mm^3^
Hemoglobin (g/dL)	35.6 g/dL
Platelet count (per μL)	219.000/μL
Viral load (copies/mL)	Undetectable
Anti-HCV antibodies	Detected

## Data Availability

Not applicable.

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
