# Peer review of "HCV-HIV Chronic Coinfection Prevalence in Amazon Region"

_jcm, 2022, doi:10.3390/jcm11247284_

Round 1

Reviewer 1 Report

This is a study on the prevalence of HIV/HCV co-infection among people living with HIV in Belem, State of Para, Brazil. The strengths of the study included: its prospective nature and the relatively large sample size. Interestingly, only one patient was found to have HCV by PCR. 

My questions/comments are as follows:

-- Line 40: suggest to delete "each of" (viral hepatitis causes about 1.34 million deaths worldwide; this is the total count and stating 'each of' might erroneously imply a higher attributable mortality)

-- Line 61: Please clarify and include a more appropriate citation to support the statement 'genotype 2... is due to the trans-Atlantic slave trade."

-- Line 47: please include the definition of BMH on first use (Brazilian Ministry of Health?)

-- How many patients reported injecting drug use? Please clarify this in the Results section and the Abstract.

 -- Include in the limitations section the possibility of missing hepatitis C infection during the window period prior to antibodies being detected by the screening test used, since HCV polymerase chain reaction was not performed on all blood samples

-- Please clarify 'genotype 2b' in line 184. 

Author Response

Journal of Clinical Medicine

Please find enclosed the revised manuscript (Manuscript ID: jcm-2027572) entitled “HCV-HIV chronic coinfection prevalence in Amazon region”.

We were pleased to see that the reviewers suggested some revisions of the manuscript.  All corrections made are highlighted in the manuscript and the authors thanks the reviewer for the interesting suggestions, which have improved the manuscript.

We have included this reply to reviewer's concerns comments below and hope the manuscript will be suitable for publication in Journal of Clinical Medicine – section: Infectious Diseases – Special Issue: Chronic HCV Infection: Clinical Advances and Eradication Perspectives - Part II.

After reading both reviewers similar comments we authors highlighted the answers to reviewer 1in green and we made all the changes that were proposed. Regarding reviewer 2we authors highlighted the answers in green and we made all the changes that were proposed, we made in the paper to demonstrate the accepted reviewers suggestions to improve the manuscript and increase paper acceptance.

Reviewer 2 Report

Line 38-42: please provide the newest data, not from 2017. I also don’t understand why the reference 2 was used regarding the sentence about WHO report

Line 47: BMH was used for the first time – you should explain (please provide full name)

Line 71 – not agree, but agreement

Line 84 – not PLWHA but PLWH

In the section “Results” I would suggest to do a table with patients’ baseline characteristics

I would suggest to add some infomation in the discussion like about prevalence of HCV in general population in the Belem city and the access to antivirals in Brazil

Author Response

(The authors gave the same response as above.)
